# Microstructure and Mechanical and Impact Behaviors of WC-Particle-Reinforced Nickel-Based Alloy Surfacing Layers at Evaluated Temperatures

Li Zhang [1,2], Shengli Li [1,*], Chunlin Zhang [1], Xingang Ai [1] and Zhiwen Xie [3,*]

1 College of Materials and Metallurgy, University of Science and Technology Liaoning, Anshan 114051, China; zhangli_198803@126.com (L.Z.); zhangchunlin1994@163.com (C.Z.); aixingang@126.com (X.A.)
2 College of Materials Science and Engineering, Yingkou Institute of Technology, Yingkou 115004, China
3 College of Mechanical Engineering and Automation, University of Science and Technology Liaoning, Anshan 114051, China
* Correspondence: lishengli@ustl.edu.cn (S.L.); xzwustl@126.com (Z.X.)

**Abstract:** A WC-particle-reinforced nickel-based alloy surfacing layer was fabricated on 42CrMo ultra-high-strength steel. The microstructure and the mechanical and impact-damage behaviors of the surfacing layers at the evaluated temperatures were investigated using X-ray diffraction (XRD), scanning electron microscopy (SEM), energy-dispersive X-ray spectroscopy (EDX) and the Vickers hardness tester. Results showed that these WC particles gradually changed from elongated and crisscross needle-like phases to blocks with the increase in impact temperature. Numerous carbide phases (e.g., $(Cr,Ni,Fe)_{23}C_6$) and $\gamma$-Ni phases were formed in the substrate matrix. The surfacing layer showed a typical brittle fracture, and the impact energy decreased with the increase in temperature. Moreover, the surfacing layer showed a clear quasi-cleavage fracture morphology without dimples after a 600 °C impact test but exhibited a mixture of dimple fractures and cleavage fractures after the 200 °C and 400 °C impact tests. The Vickers fracture toughness test showed that the average hardness of the surfacing layer after a 600 °C impact test was 383 $HV_{1.0}$, which is about 0.8 times that after the 200 °C impact test. In addition, the WC particles in the surfacing layer after the 600 °C impact test showed the highest fracture toughness, but the corresponding Ni40A binder phase possessed the lowest fracture toughness.

**Keywords:** plasma arc surfacing; WC doping; Charpy impact; fracture toughness; hardness

## 1. Introduction

The descaling roller is an important part of hot rolling and also the most damaged part of the rolling process. The consumption or destruction of the descaling roller will increase the maintenance required for equipment or cause downtime, which not only affects the production cost and productivity but also affects the surface quality of the rolled materials. In recent years, researchers have been paying attention to the quality and wear resistance of rolls [1], but there has been little research on the descaling roller. The descaling roller, especially the descaling roller in the high-pressure water tank behind the heating furnace, has a bad working environment, and it must bear the large impact force of the high-temperature steel billet, the thermal stress caused by the temperature change and the shear stress caused by the friction on the surface, so cracks and spalling often occur [2]. In order to improve the life of the descaling roller and reduce the cost, we actively seek suitable repair methods.

There are many techniques for preparing metal matrix composite coatings, such as surface welding [3,4], laser cladding [5,6] and plasma surfacing [7,8]. Among the many preparation methods, plasma surfacing technology has the advantages of low cost, convenient maintenance and strong material adaptability. In addition, as its heating and

cooling rate is lower than that of laser cladding, the molten pool can be maintained for a long time, which is conducive to the formation of a homogeneous structure and greatly reduces defects [9,10]. As a metal-based composite coating of the hard phase, tungsten carbide has been widely used in the industry due to its high wear resistance, good hardness and satisfactory toughness [11–13]. In addition, tungsten carbide has good toughness, strength and wear resistance due to the comprehensive effect of bonding and is widely used in wear-resistant tools [14,15].

There are many studies on carbide-particle-reinforced nickel-based coatings, and most scientists focus on the influence of WC content change on its wear resistance and corrosion resistance. Plasma surfacing technology is used to prepare the surfacing layer on a steel plate. By changing the WC content, it was found that the wear resistance is the best when the WC content is 60%. The main reason for this is that the $Fe_3W_3C$ carbide phase in the coating improves the hardness and wear resistance of the coating [16]. The Ni–WC composite coating was prepared on an EH40 steel plate, and the influence of temperature change on the tribological properties of the Ni–WC coating was analyzed. The results showed that Ni + 15%WC showed excellent wear resistance at $-20°C$, and its wear volume and wear rate were minimal, mainly due to the presence of WC, which reduced the plastic deformation caused by the friction ball and the coating surface. The wear mechanism changes from adhesive wear to abrasive wear [17]. In order to improve the corrosion resistance of the Ni coating, a directional-structure Ni60 coating was prepared on S45C steel [18]. The mechanical properties and fracture behavior of the WC/Ni coatings depend on their content of WC-reinforced particles. With the increase in WC particle content, the tensile property decreases [19]. WC/Ni composite coatings have a higher hardness, a lower friction coefficient and lower wear [20]. After adding WC/Ni particles [21], the microhardness and wear resistance of the Ni-based alloy coating are significantly improved. Adding WC particles is an effective method to improve the tribological properties of conventional Ni-based alloy coatings.

It can be seen from the above results that the WC/Ni coating has important research value, but most of the current research stays in the flat surfacing, and research on the high-temperature Charpy impact performance of the coating, which is of great significance for the actual use environment of the descaling roller, is still insufficient. Charpy impact tests were originally developed for metals and later extended to non-reinforced plastics. This test seems to be a reasonable choice for metal matrix composite coatings. Although there are uncertainties in the measurement of energy dissipation due to the kinetic energy of the broken specimen and the multiple fractures and delamination, impact energy does provide a useful measurement.

In this work, the WC/Ni40A coating was surfaced on a $\varphi$150mm 42CrMo steel matrix using plasma surfacing technology. The Charpy impact behavior of the WC-particle-reinforced nickel-based double-gradient surfacing layer at different temperatures was then evaluated based on phase content, microstructure and fracture morphology.

## 2. Materials and Methods

A surfacing layer was prepared using a 42CrMo roller with a diameter of $\varphi$150mm as the base material. It was characterized by higher strength and good toughness, as well as no obvious tempering brittleness and very good hardenability. Furthermore, the steel showed very good low-temperature impact toughness and a higher fatigue limit after tempering treatment. The chemical composition of the base steel as per the manufacturer is listed in Table 1.

**Table 1.** Chemical composition of the base material, 42CrMo (wt%).

| C | Mn | Si | Cr | Mo | Ni | V | Cu | P | S | Fe |
|------|------|------|------|------|------|------|------|-------|-------|------|
| 0.42 | 0.55 | 0.26 | 0.98 | 0.16 | 0.02 | 0.01 | 0.01 | 0.011 | 0.015 | Base |

The duplex surfacing layer consisted of a transition layer and a hard-standing, in which the powder of the hard-standing was made of a hard phase (spherical tungsten carbide) and a bonding phase (nickel-based alloy Ni40A). Figure 1 shows the spherical tungsten carbide and point scan components. The spherical tungsten carbide had complete sphericity, a smooth surface and no obvious defects. The particle size of the sphere-cast tungsten carbide was about 100~200 mesh. The transition layer was a Ni-based alloy Ni40A powder. The average hardness of the spherical tungsten carbide and the Ni-based alloy powder was 2200 $HV_{0.1}$ and 35 HRC, respectively.

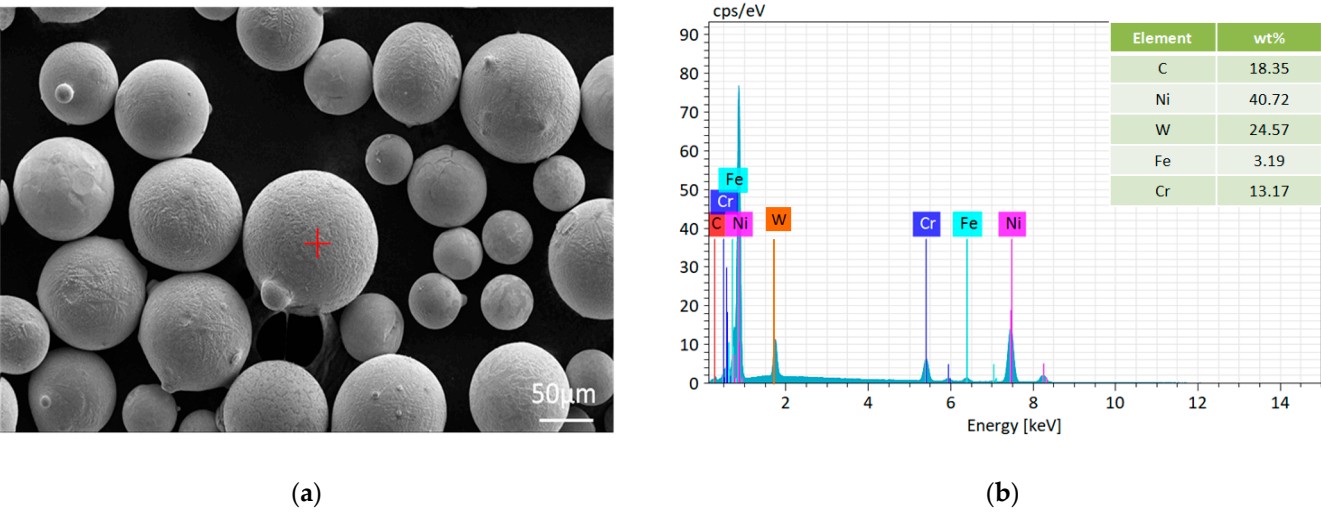

| Element | wt% |
|---------|-------|
| C | 18.35 |
| Ni | 40.72 |
| W | 24.57 |
| Fe | 3.19 |
| Cr | 13.17 |

(**a**)　　　　　　　　　　　　　　　　　　(**b**)

**Figure 1.** Tungsten carbide powder and EDS point-scanning analysis. (**a**) Tungsten carbide, (**b**) EDS point scan analysis marked in red in (**a**).

For the hardfacing, Ni40A and spherical tungsten carbide powders were fused on the base material, 42CrMo, with a plasma hardfacing machine, DML–V03BD. The machine enables pulse hardfacing with a current of 2–300 A and a frequency of 0–50 Hz. We applied additional material to the base material using the hardfacing machine. Before proceeding with the hardfacing process itself, the surface of the base material was prepared properly. Then, the base material was preheated to 550 °C and the surfacing process was initiated in the self-designed heating furnace (Figure 2), after which the furnace was cooled to room temperature. The hardfacing parameters for the layers are shown in Table 2.

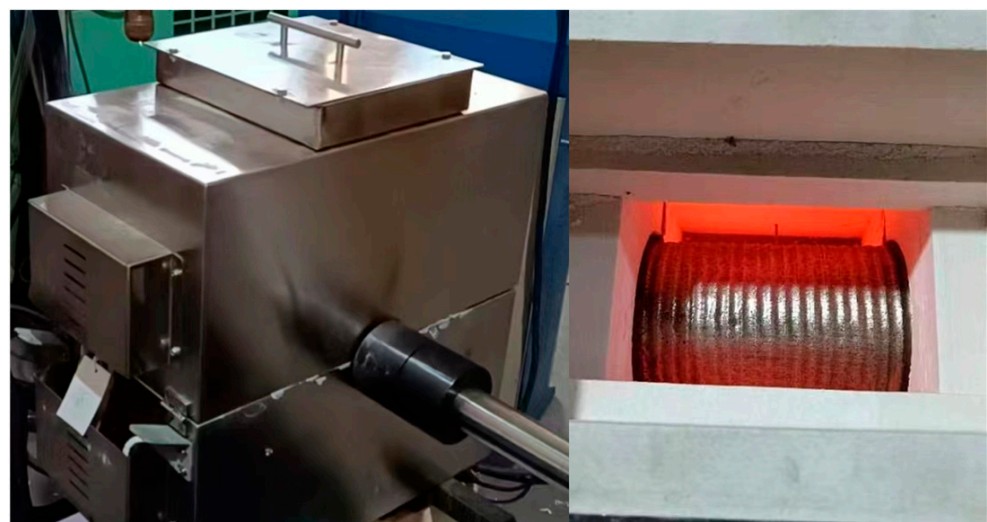

**Figure 2.** Self-designed heating furnace.

**Table 2.** Hardfacing parameters for duplex surfacing layers.

| Layers | |
| --- | --- |
| Pulsating direct current: | 140 A, with a frequency of 50 Hz |
| Trajectory: | swing step 20 mm |
| | swing length 471 mm |
| Serving: | 42 g·min$^{-1}$ |

The Charpy impact test was carried out on a pendulum impact tester at 200 °C, 400 °C and 600 °C, respectively. Each sample was heated at a rate of 10 °C/min, and the impact test was carried out after reaching the target temperature for 5 min. The Charpy standard V-notch impact specimen with a gauge size of 10 mm × 10 mm × 55 mm was prepared along the axis of the descaling roller. The preparation of impact sample is shown in Figure 3. After surfacing, the Charpy impact behavior of the WC-particle-reinforced nickel-based duplex surfacing layer at different temperatures was evaluated from the phase content, microstructure and fracture morphology.

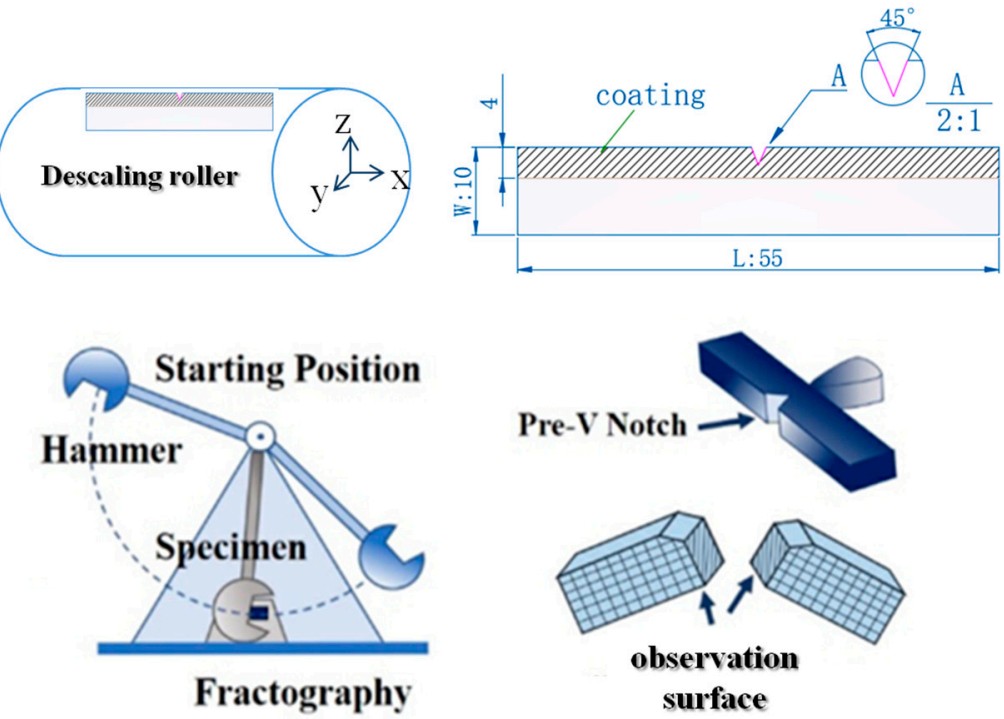

**Figure 3.** Charpy shock sample size and preparation process.

The microstructure and fracture mechanism of different regions (fusion zone, heat affected zone and base metal) were analyzed in detail; especially, the mixing mode of the additives and the matrix materials were determined. It was necessary to use the combined methods of light microscopy (LM) and scanning electron microscopy (SEM). After polishing, the samples were etched with aqua regia, and the microstructure was observed. Metallographic observation at a magnification of 100X was carried out with a Zeiss Primotech optical microscope. The morphological details of the microstructural constituents were revealed using SEM (Zeiss, Sigma 500, Jena, Germany), with analytical units, and energy-dispersive X-ray spectroscopy (EDX; Bruker XFlash 6-30, Billerica, MA, USA) using back-scattered electrons (BSEs). To determine the chemical composition of the structural phases and for the overall chemical elemental analysis, energy-dispersive X-ray spectroscopy (EDX) was used. The EDX analysis presented in this work should be considered a semi-quantitative analysis. Observations were performed on nital-etched samples at an accelerating voltage of 15 kV. The phase structure of the samples was

examined using an X-ray diffractometer (XRD, D8 ADVANCE) with monochromatic Cu Kα radiation. The scanning time and range were 2 min and 5°–100°, respectively. Hardness measurements were carried out across different areas using the HV-1000 microhardness tester, by means of the Vickers method according to EN ISO 6507-1. A load of 9.8 N (1000 g, $HV_{1.0}$) and a full-load dwell time of 15 s were used for the test. On each metallographic section, the hardness was measured in three lines from the weld surface to the base material. On the basis of Vickers indentation, the average values of ten indentation cracks were taken to calculate the fracture toughness of the WC particles and the nickel base.

## 3. Results and Discussion

### 3.1. Microstructure Characterization of the Fabricated Surfacing Layer

Figure 4 is the XRD pattern of the original WC-reinforced nickel-based alloy surfacing layer. The results show that the main phases in the surfacing layer are γ-Ni solid solution, $M_{23}C_6$(M:Cr,Ni,Fe)-type interstitial compound, $Cr_7Ni_3$ phase, WC and $W_2C$. After plasma surfacing of the Ni40A and WC powders, the C element in the surfacing layer reacts with Cr and Ni in the molten pool, and the high-hardness $(Cr,Ni,Fe)_{23}C_6$ and $Cr_7 Ni_3$ phases are precipitated from the liquid metal. The formation of the $W_2C$ phase is due to the decomposition of WC into C and $W_2C$ at high temperature [22].

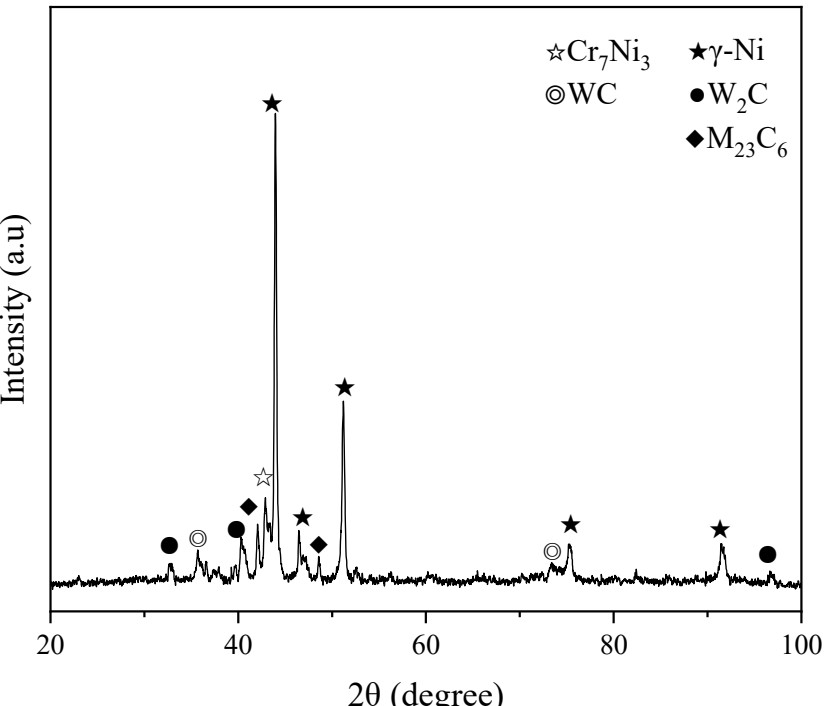

**Figure 4.** XRD patterns of the fabricated samples.

Figure 5 shows the microstructure of the hard surface layer and the transition layer of the original preparation state. Figure 5b,d is the bonding zone and transition layer structure after surfacing. From Figure 5b, it can be seen that there are no defects in the bonding zone of the original preparation state, showing good metallurgical bonding, and that the bonding strength reaches 467 MPa. From Figure 5d, it can be clearly seen that the transition layer is dominated by dendrites. Figure 5a,c is the microstructure of the hard layer after surfacing. The addition of WC particles in the hard surface layer inhibits the growth of grains in the surfacing layer, so as to achieve the purpose of grain refinement. WC with a smaller particle size decomposes after heating to form $W_2C$, while WC particles with larger particle size decompose less, so there are relatively more WC phases and γ-Ni solid solutions in the prepared state.

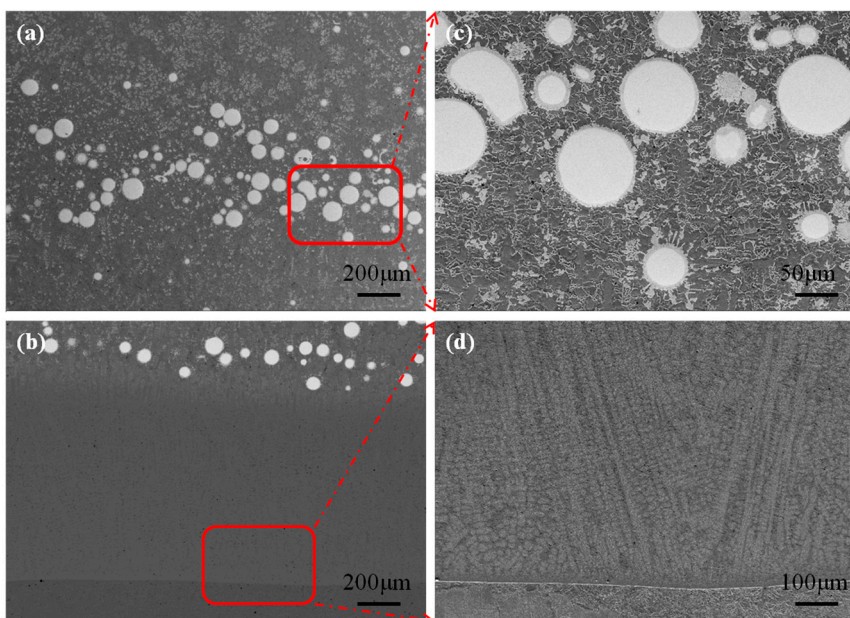

**Figure 5.** SEM images of the fabricated sample: (**a**) hard-standing, (**b**) transition layer, (**c**,**d**) enlarged figures corresponding to the red region in (**a**,**b**).

### 3.2. Charpy Impact Behavior of the Surfacing Layer at the Evaluated Temperatures

With the increase in temperature, the Charpy-impact absorbed energy of the specimen gradually decreases, and the decrease is serious at 600 °C, as shown in Figure 6. Thus, it can be concluded that the external energy required for the same specimen to fracture under the same load is small, and the corresponding fracture-resistance ability is weak, and the corresponding fracture toughness value should be relatively small.

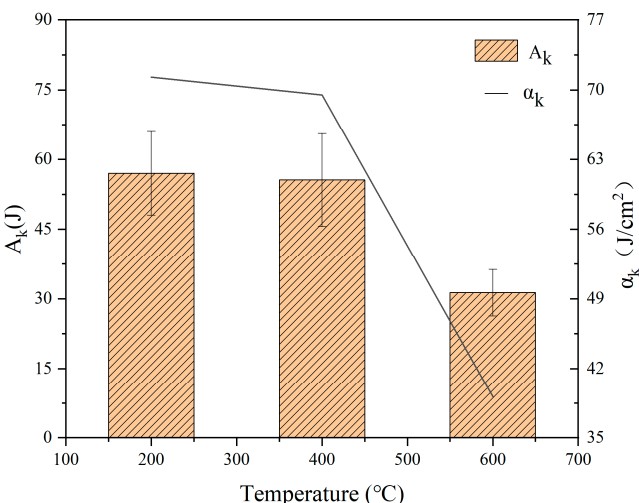

**Figure 6.** Charpy impact test data at different temperatures.

The fracture mechanisms of the 200 °C, 400 °C and 600 °C impact samples under dynamic impact loads were analyzed by observing the microscopic morphological characteristics. The SEM fracture morphology of the sample after high-temperature Charpy impact tests at 200 °C, 400 °C and 600 °C is shown in Figures 7 and 8. It can be seen from the macroscopic morphology of the fracture that the impact fracture at 600 °C has obvious delamination, while the delamination of the impact specimens at 200 °C and 400 °C is relatively small. The separation area is mainly at the planar crystal position, which corresponds

to the microscopic metallographic phase. It shows that planar crystals in large areas will affect the bonding strength of the overlay layer [23].

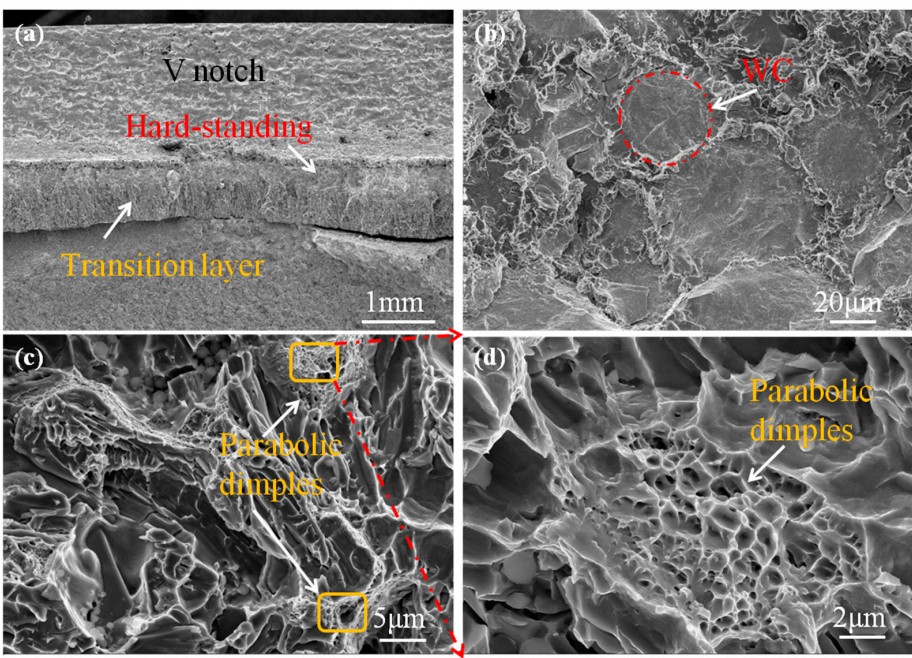

**Figure 7.** Microfracture morphology of impact specimen at 200 °C: (**a**) fracture enlarged diagram, (**b**) microstructure of hard-standing, (**c**) microstructure of transition layer, (**d**) enlargement of the parabolic dimples in (**c**).

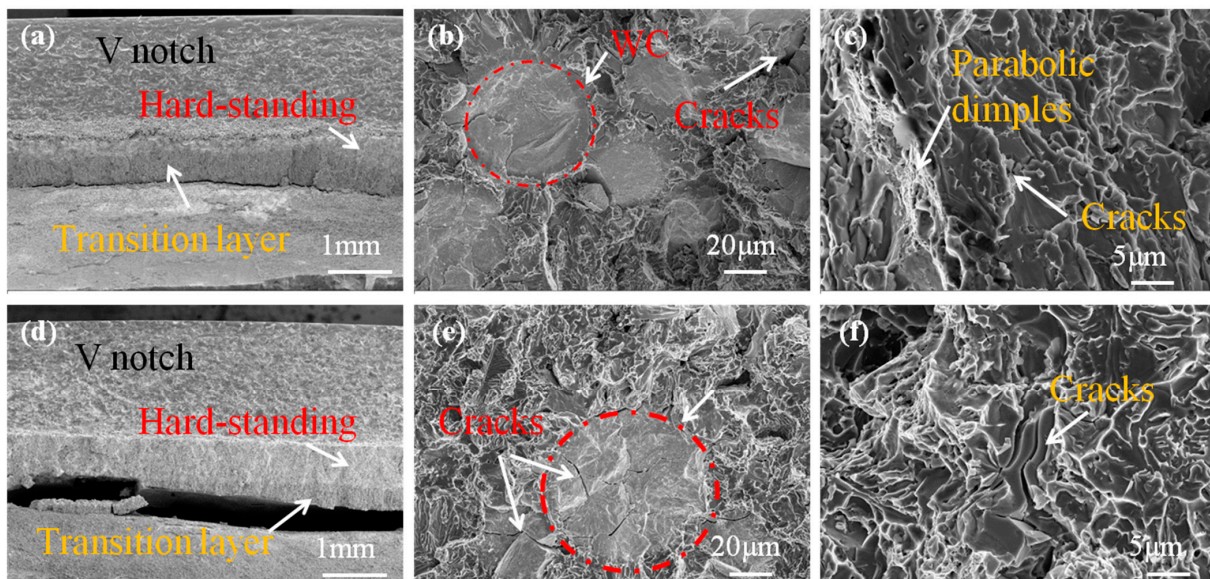

**Figure 8.** Microfracture morphology characteristics of samples: (**a**–**c**) 400 °C; (**d**–**f**) 600 °C.

For the 600 °C impact, the fracture surface of the transition layer of the specimen is mainly composed of a small cleavage surface. This cleavage surface is connected by small, curved tear edges without tendons but with noticeable cracks. At the same time, there are obvious "river pattern" and cleavage steps on the large cleavage surface. The fracture characteristics indicate quasi-cleavage fracture, and there are radial steps on the surface, which are characteristics of brittle fractures [24]. There are obvious cracks on the WC particles in the hard surface layer, and the cracks extend along the $W_2C$ phase. The fractures of other nickel bases are still small cleavage surfaces, and obvious cracks

can be found, as shown in Figure 8e,f. When the impact temperature is 400 °C, a small number of ligaments and cracks exist on the fracture surface of the transition layer of the specimen, and the plastic deformation mainly occurs during crack propagation. In the same direction, the dimple of the 200 °C sample is deeper and larger than that of 400 °C sample. This shows that the plastic deformation of the 200 °C sample coating is larger than that of the 400 °C sample coating under impact load, and the plastic deformation work consumed during crack propagation is larger than that of the 400 °C sample coating, as shown in Figures 7c and 8c. When the impact temperature of the sample coating is 200 °C, the fracture morphology of the sample is composed of a large number of dimples and cleavage planes, and the dimple band is surrounded by cleavage planes and tear edges as shown in Figure 7d, and the secondary crack is not obvious, which is a quasi-cleavage fracture morphology [25]. It can be seen that the fracture mechanism of the 200 °C and 400 °C impact specimens is a mixture of dimple fractures and dissociative fractures. To sum up, the impact toughness of the 200 °C and 400 °C specimens is better than that of 600 °C impact specimen.

### 3.3. Microstructure Characterization of the Surfacing Layer after the Impact Tests

Figure 9 shows the XRD patterns of the WC–Ni hardfacing at different temperatures. The main phases included $W_2C$, WC, $M_{23}C_6$(M:Cr,Ni,Fe), $\gamma$-Ni and $Cr_7Ni_3$. The presence of $(Cr,Ni,Fe)_{23}C_6$ may be attributed to the dissolution of C and W into the binder, which allows the formation of hard phases. Similar results were also confirmed by Liyanage et al. [26]. The impact specimens at 600 °C have the maximum peak value of $(Cr,Ni,Fe)_{23}C_6$, following the 200 °C and 400 °C samples. Comparison of the height ratios of the peak of WC ($2\theta = 35.7°$) and $W_2C$ ($2\theta = 39.6°$) shows that the height ratio is nearly steady from the 200 °C sample to the 600 °C sample, and the peak of $\gamma$-Ni and $(Cr,Ni,Fe)_{23}C_6$ increases sharply from the 200 °C sample to the 600 °C sample. The large differences in the height ratios of $\gamma$-Ni of different samples may be attributed to the dissolution of Ni40A during different shock temperatures. With the increase in temperature, the surface WC decomposition is serious, and the $W_2C$ brittle phase increases. Compared with the original hardfacing layer, the WC phase content is reduced.

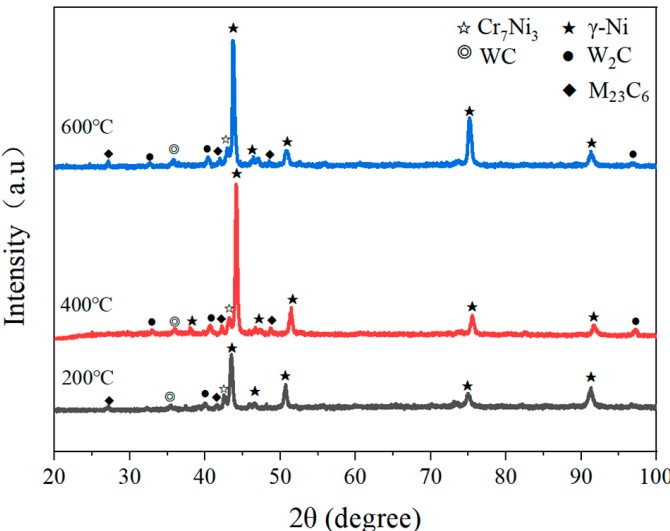

**Figure 9.** XRD patterns of hardfacings at different temperatures.

Figure 10 is the cross-sectional microstructure of the WC-reinforced Ni-based coating prepared using plasma surfacing at different Charpy impact temperatures far from the fracture area. It can be seen that the coating prepared through plasma overlay welding is well formed. There is a fusion line between the cladding layer and the substrate, indicating that the coating is metallurgically combined with the substrate. Moreover, no holes were found in the coating, consistent with the original overlay layer. The coating can be divided

into the planar crystal region, the cellular crystal region, the columnar crystal region and the equiaxed crystal region from the fusion line to the top, corresponding to the regions ①~④ in Figure 10f. Plasma surfacing has an extremely fast cooling process. The growth of the cladding layer structure is controlled by the direction of maximum heat flow loss, and the columnar crystals (or columnar dendrites) have directional solidification characteristics. Therefore, in the process of plasma surfacing, the microstructure is mainly determined by the solidification rate, R, and the temperature gradient, G [27]. In the process of plasma overlay welding, due to the rapid heat dissipation of the substrate at the bottom of the molten pool, the maximum temperature direction of the cladding layer is perpendicular to the surface of the substrate, and the temperature gradient, G, is relatively high. At the same time, the slope of the temperature gradient, G, does not intersect with the crystallization temperature curve. There is almost no component supercooling at the solid–liquid interface, and the solid–liquid interface advances extremely fast. At this time, a thin layer of planar crystal is formed at the interface between the substrate and the coating, as shown in Figure 10b,d,f. With the increase in the distance between the cladding layer and the interface, the formation of the planar crystal reduces the heat conduction velocity of the molten pool, and the temperature gradient gradually decreases. The temperature curve begins to intersect with the crystallization temperature curve, and the solidification rate of the liquid–solid interface slows down. Composition undercooling also begins to appear in the molten pool. At this time, cellular crystals and columnar crystals are gradually formed from the planar crystal area in the coating to the upper part of the coating, as shown in Figure 10a,c,e.

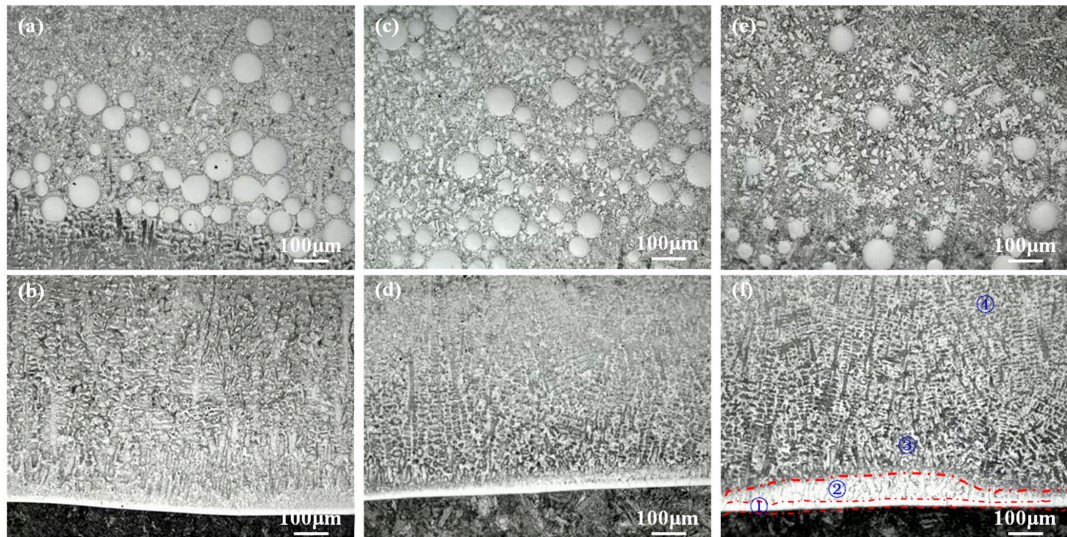

**Figure 10.** Cross-sectional optical micrographs of the samples at different impact temperatures: (**a**) and (**b**) 200 °C, (**c**), and (**d**) 400 °C, (**e**) and (**f**) 600 °C.

After the high-temperature impact operation, due to the influence of the working temperature and the impact force, the microstructure of the transition layer and the hard surface layer becomes smaller, and the plane crystal area of the 600 °C fusion zone becomes larger, as shown in Figure 10f. The higher the impact temperature, the greater the dissolution of the WC particles, as shown in Figure 11a–c. The overall WC particle size is lesser than that at 200 °C and 400 °C. In addition, the excess carbon produced by WC dissolution diffuses into the metal matrix to form more carbide phases $(Cr,Ni,Fe)_{23}C_6$. Liyanage et al. [26] report that the Ni content increases the dissolution of WC particles and promotes the formation of $\gamma$-Ni and carbide phases in the metal matrix, which is consistent with the results shown in Figure 9.

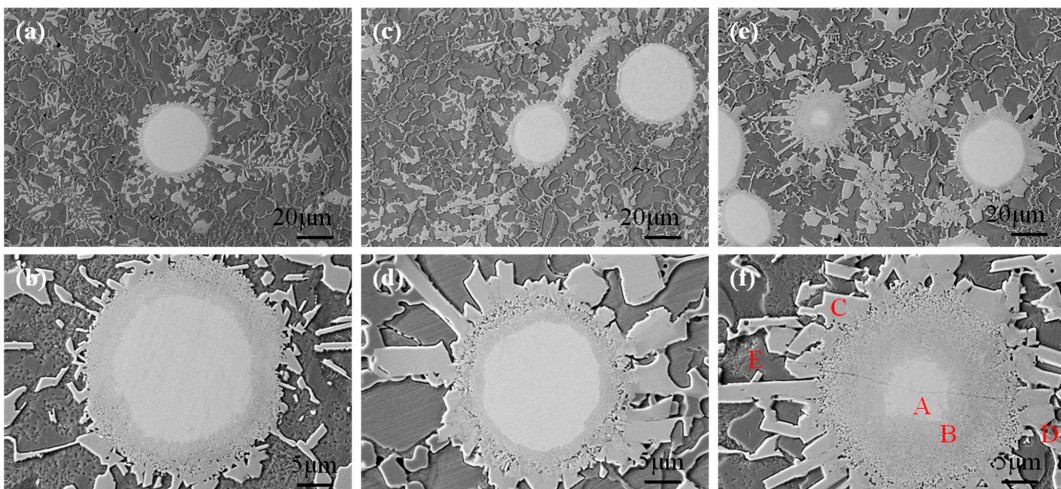

**Figure 11.** SEM images of the upper cladding layer at different impact temperatures: (**a**,**b**) 200 °C, (**c**,**d**) 400 °C, (**e**,**f**) 600 °C.

When the impact temperature is 200 °C, the WC and Ni40A matrices diffuse with each other. WC particles gradually decompose and form a narrow transition layer, forming a small number of thin, strip-like structures at the edge of the WC particles. Furthermore, precipitation begins around the transition layer, as shown in Figure 11a,b. When the shock temperature rises to 400 °C, a massive structure precipitates around the transition layer. The WC particles form massive carbides at the edge, and separation begins, as shown in Figure 11c,d. When the impact temperature is 600 °C, the block structure of the transition layer increases significantly, and the dissolution of WC particles becomes obvious. Cracks appear inside and in between adjacent WC particles, as shown in Figure 11e. The appearance of cracks is mainly due to the increase in the carbide volume fraction and the decrease in the binder phase with the increase in WC particle dissolution. The thermal expansion coefficients of WC particles and Ni40A are different. Under the condition of heating and cooling, the thermal stress produced by the cladding exceeds the strength limit of the Ni40A matrix, which results in cracks. With the increase in impact temperature, the degree of WC decomposition increases, and the nickel-based structure becomes coarse. This is mainly due to the dissolution of WC into bulk $W_2C$ [28], which occupies a large area and reduces the nickel-based structure of the same area.

The amount of these carbides keeps increasing with an increase in the impact temperature and is accompanied by a significant increase in the dissolution of WC particles in the matrix. EDS analysis of the carbide phases in samples impacted at 600 °C (Figure 11f) shows a sharp decrease in the concentration of W from 47 wt% to 18 wt% (Table 3—points A and C). It is shown that this is due to the high solubility limit of W in Ni at these temperatures and the dissolution of WC particles, consistent with the W–Ni binary phase diagram [29]. Therefore, the amount of WC decarburized in the matrix depends on the selected impact temperature. When the impact temperature is 600 °C, cracks appear on the surface of some WC particles, as shown in Figure 11f, which also promotes the rapid dissolution of WC, so that the internal composition of the particles changes gradient, as shown in Figure 12.

**Table 3.** EDS analyses for points A, B, C, D and E illustrated in Figure 11f.

| At.% | C | Cr | Fe | Ni | W |
|------|-------|-------|-------|-------|-------|
| A | 52.54 | - | - | - | 47.46 |
| B | 28.38 | 12.75 | 10.55 | 21.85 | 26.47 |
| C | 29.65 | 18.02 | 8.23 | 25.90 | 18.20 |
| D | 17.01 | 3.80 | 20.36 | 58.83 | - |
| E | 12.46 | 6.75 | 26.57 | 54.21 | - |

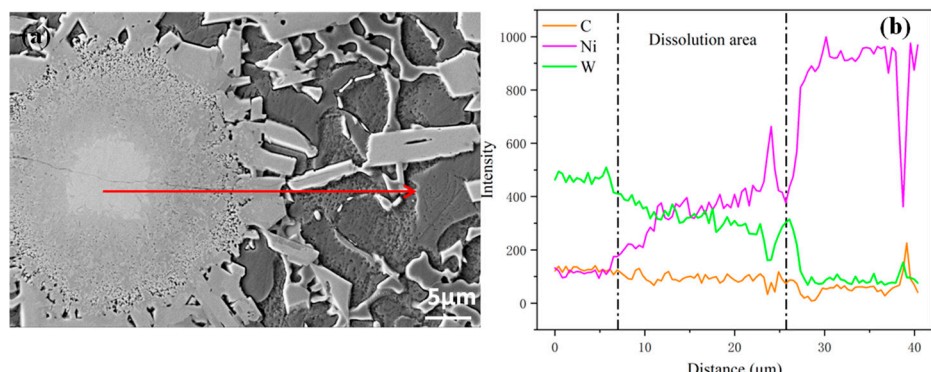

**Figure 12.** EDS cross-sectional scanning results of WC particle at 600 °C: (**a**) Microtopography of individual WC particles, (**b**) Corresponding to the distribution of elements of the red line in (**a**).

Figure 13a,b shows the surface scanning image and needle phase of a single WC particle after impact at 400 °C. Energy-dispersive X-ray spectroscopy (EDS) analysis of the particles shows high concentrations of Ni (34.40 wt%) and W (34.92 wt%) and low concentrations of Cr (5.89 wt%). The dissolution pattern of WC is as follows: the center of the WC remains unchanged; first, the edges begin to dissolve, and then many elongated and crisscrossing needle-like phases are precipitated, as shown in the red area in the figure. The closer to the nickel matrix, the looser the distribution of needle-like phases, and a small amount of granular material appears. The composition change area from the center to the edge of the WC particles is consistent with Figure 13c. The mass fraction of W in the composition decreases and the mass fraction of Ni increases, which indicates that W, C in WC and elements in the nickel matrix diffuse into each other. In addition, Cr, Fe and W react with C to form compounds mainly composed of $(Cr,Ni,Fe)_{23}C_6$, WC and $W_2C$.

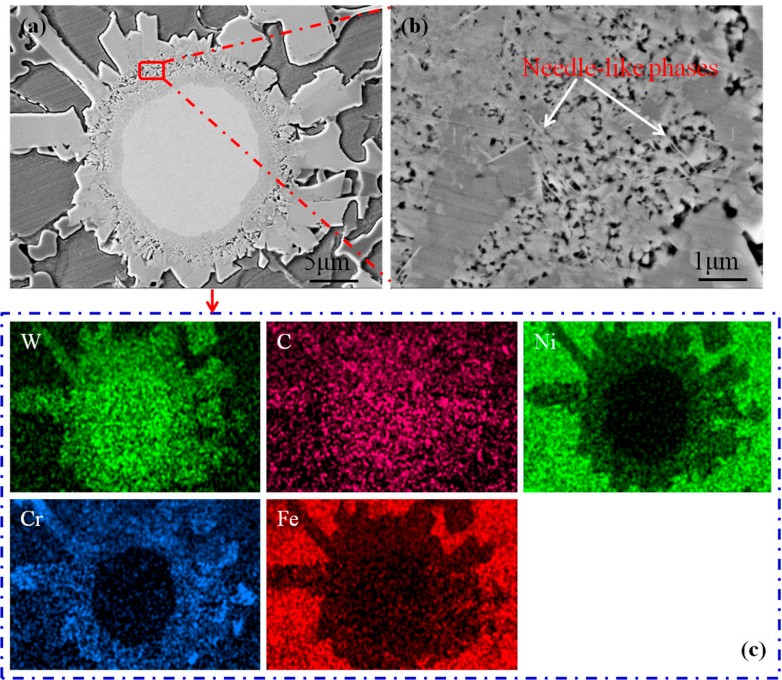

**Figure 13.** SEM images and corresponding element mappings after 400 °C impact test: (**a**) Microtopography of individual WC particles, (**b**) Corresponds to an enlarged view of the red area in (**a**), (**c**) Distribution of elements corresponding to (**a**).

### 3.4. Effect of Impact Temperature on the Mechanical Behavior of the Coatings

The cross-section morphology at 600 °C and the Vickers hardness curve at different temperatures are shown in Figure 14. The cross-section is obviously divided into four regions, namely, substrate, heat-affected zone, transition layer and hard-standing. For the 600 °C impact, the macroscopic fracture separates from the bonding layer of the overlay layer. The hardness curve defines the interface of the matrix as the initial point. The microhardness increases sharply when entering the heat-affected zone. Due to the formation of new grains, the alloy has a higher stress concentration, which increases the microhardness. When the hardness of the transition layer into Ni40A decreases, the microhardness of the hard-standing increases with the increase in the hard phase content, and the hardness trend of three different temperature impact samples is almost the same. The hardness of the substrate and the heat-affected zone is not affected by the impact temperature. The hardness of the transition layer and the hard-standing varies greatly: especially when the impact temperature is 600 °C, the average hardness of 383 $HV_{1.0}$ is about 0.8 times that at the impact temperature of 200 °C. The main reason is that cracks appear inside the particles at 600 °C, and the decarburization of WC particles is serious. The decarburization of WC particles is the main factor affecting the coating hardness. The hardness improvement of the heat-affected zone is mainly related to the extremely fast heating and cooling rate of the surfacing process, which promotes the martensitic transformation of the matrix structure. Compared with the hardness of the as-prepared state, the hardness decreases slightly, which is not compared in this paper. The hardness of the transition layer decreases to a certain extent. The main reason is that more Fe elements are dissolved in this area due to the dilution of the matrix, which reduces the content of solute atoms and the content of hard carbide precipitates.

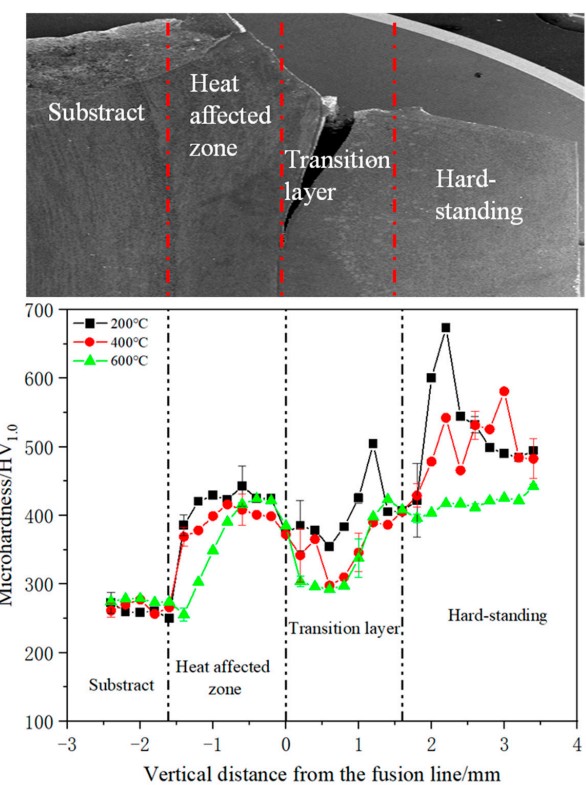

**Figure 14.** The microhardness distribution along the cross-sectional depth.

### 3.5. Fracture Toughness of the Surfacing Layer after Impact Tests

The fracture toughness of the surfacing layer at different impact temperatures was evaluated using Vickers indentation. The load was 1 kg and the load holding time was 15 s. The fracture toughness (KIC) is calculated using the formula KIC = 0.0028 $(HP/L)^{1/2}$ [30]

after measuring the crack length, where H is the indentation hardness, P is the indentation load and L is the total crack length. Ten hardness points were measured on WC particles and Ni40A, respectively, and their average values were determined. In Figure 15, (a–f) represent the Vickers hardness points in WC particles and nickel-based binder phases at different impact temperatures, where (a, b) is 200°C, (c, d) is 400°C and (e, f) is 600°C. The change in the average toughness value (KIC) with the impact temperature is shown in Figure 16. The fracture toughness of WC particles at 600 °C is higher than that at 200 °C and 400 °C, and the average crack length on the particles is relatively short, which is related to the degree of dissolution of WC particles. With the increase in impact temperature, the synergistic effect of the Ni binder and WC grain is conducive to releasing local stress due to the effect of thermal stress, thus improving the fracture toughness of the cemented carbide [31]. However, the fracture toughness of the Ni40A binder at 600 °C is poor, mainly due to the serious decomposition of the WC particles, which increases the hardness in the binder, and thus makes its fracture toughness lower than that at 200 °C and 400 °C. However, there are few cracks in the binder of the overall Ni40A.

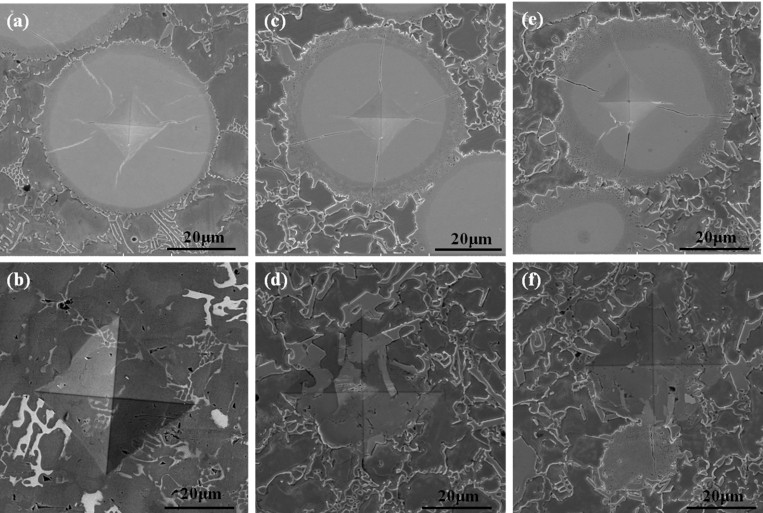

**Figure 15.** Vickers indentation images of the WC particles and nickel-based matrix at different impact temperatures: (**a**,**b**) 200 °C, (**c**,**d**) 400 °C, (**e**,**f**) 600 °C.

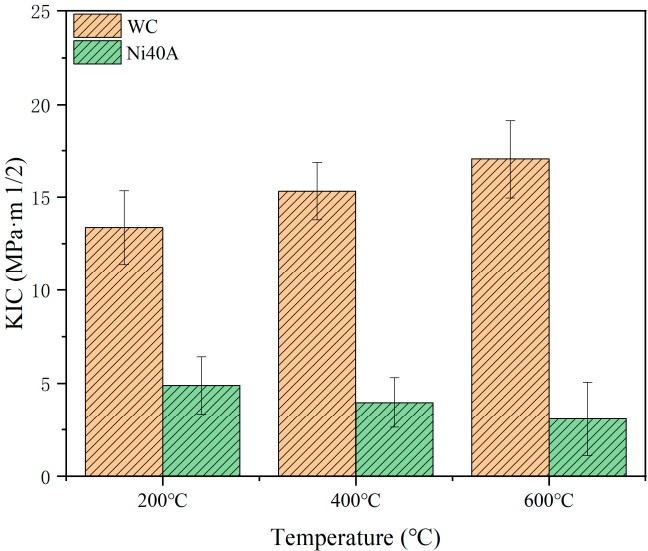

**Figure 16.** The fracture toughness of Ni40A and WC on the surface of impact samples.

## 4. Conclusions

This study investigated the microstructure and the mechanical and impact behaviors of the surfacing layer at the evaluated temperatures. The WC particles' edge morphology was very sensitive to the impact temperature, changing from an elongated, intersecting, needle-like phase to a massive phase with the increase in impact temperature. Plenty of $(Cr,Ni,Fe)_{23}C_6$ and $\gamma$-Ni phases also formed in the alloy matrix. A typical brittle fracture feature was characterized in these tested samples, but the impact energy decreased with the increase in impact temperature. Moreover, a quasi-cleavage fracture pattern without depression was characterized in the surfacing layer after the 600 °C impact test, but this surfacing layer exhibited a mixed fracture pattern of dimple fractures and dissociative fractures after the 200 °C and 400 °C impact tests. In addition, the average cross-section hardness of the sample was about 383 $HV_{1.0}$ at an impact temperature of 600 °C, which was lower than that at an impact temperature of 200 °C. Moreover, at the impact temperature of 600 °C, the WC particles possessed the highest fracture toughness, but the lowest fracture toughness was detected in the Ni40A bonding phase.

**Author Contributions:** Conceptualization, S.L. and X.A.; data curation, L.Z. and Z.X.; funding acquisition, S.L.; writing—original draft preparation, L.Z.; project administration, S.L. and Z.X.; writing—review and editing, C.Z. and Z.X.; supervision, X.A. and Z.X.; validation, L.Z. and C.Z. All authors have read and agreed to the published version of the manuscript.

**Funding:** This research was supported by the National Key Research and Development Program (No. 2021YFB3702004).

**Institutional Review Board Statement:** Not applicable.

**Informed Consent Statement:** Not applicable.

**Data Availability Statement:** The data presented in this study are available on request from the corresponding author. The data are not publicly available due to the size.

**Conflicts of Interest:** The authors declare no conflict of interest.

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
