# Peer review of "Microstructure and Mechanical and Impact Behaviors of WC-Particle-Reinforced Nickel-Based Alloy Surfacing Layers at Evaluated Temperatures"

_metals, doi:10.3390/met13050961_

Round 1

Reviewer 1 Report

The work contains original research results. Based on their research, the authors correctly determined the microstructure, phase composition, chemical composition, and mechanical properties of the WC particle-reinforced nickel-based alloy duplex surfacing layer. Nevertheless, the work contains errors and ambiguities that should be corrected or clarified.

After correcting and completing, I recommend the work for publication in the Metals.

Major comments:

ï‚· In the Palmqvist equation KIC=0.0028(HP/L)1/2, 1/2 should be the exponent.

ï‚· Fig. 16 shows that the hardness of WC carbides is more than 5 times higher than the hardness of the austenitic carbide matrix. The fracture toughness KIc of the WC carbide shown in Fig.17 is nearly 3 times higher than the matrix fracture. That means that the cracks of Plamquist in the matrix should be almost 2 times longer than the cracks in the carbide. There are no such cracks in the microstructures shown in Fig.16.

Minor comments:

ï‚· The title of the work seems to be narrow in relation to the content, it contains word mistakes.

ï‚· The following should be added to the description of the research methodology:

- the heating time for each test temperature,

- the source of information about the chemical composition of the substrate,

- XRD testing parameters.

ï‚· The hardness increase in HAZ independent of the heating temperature should be explained in the discussion of the test results.

ï‚· In Fig.16, the magnification marker is incorrectly given

Reviewer 2 Report

Comments to the paper of Li Zhang, Shengli Li, Chunlin Zhang, Xingang Ai, Zhiwen Xie «Charpy impact behavior of WC particle reinforced nickel based alloy duplex surfacing layer at evaluated temperature»

The paper is devoted to solving an important practical problem and is recommended for publication, taking into account the following remarks, which mainly relate to the description of the features of fracture surfaces.

 • It is important to present a diagram of the surface layer indicating the thickness of all its components and indicate in which layer the notch ends;

• (lines 193-194) "From the fracture morphology, it can be seen that the impact specimens at 200℃ and 400℃ do not have obvious cracking..". However, in Fig. 8 b shows obvious cracks with a large opening;

• Magnifications in Fig. 8e, f and Fig. 11 are hard to see;

• (206) “…no obvious cracks were found, as shown in Fig. 9(e), (f)". In these figures, the authors indicate numerous cracks;

• (211-213) “This shows that the plastic deformation of the coating sample at 200℃ is greater than the plastic deformation of the coating sample at 400℃ under impact loading, … as shown in Figure 8(c) and Figure 9(c).” It is difficult to compare these fractograms, since they were taken at different magnifications;

• What does the formation of the "parabolic dimples" shown in all the figures indicate?

Reviewer 3 Report

A very interesting article, written correctly, but I have a few observations that I would like to bring to the attention of the authors.

- it is better to use the word scientist than scholars (line 54)

- In Figure 1, there does not have to be a spherical powder, just write the morphology of the WC powder, as the shape is described in the text 

- The device in Figure 2 has no description according to the publisher's guidelines

- WC carbide is a fairly stable carbide, but in contact with steel, I tend to dissolve at temperatures above 1100 degrees Celsius. It then forms a brittle envelope, having the character of a bedded M6C carbide, into which elements from the surrounding steel also diffuse.  There is some literature data indicating this.

- What elements form M23C6 carbide?

- conclusions are too general.

- Please also note that the brittleness test result can be influenced by the substrate, in steels the brittleness is known to be in the range 450 ÷ 550 °C. The susceptibility of steel to reversible embrittlement depends on the chemical composition. Manganese, phosphorus, and trace admixtures such as antimony, arsenic, etc., most strongly increase this susceptibility. Chromium has a similar effect but to a lesser extent. If manganese, silicon, or nickel are present with the chromium, the propensity increases. Increased phosphorus content also has an influence on this brittleness.
